

# Biosynthesis of zinc oxide nanoparticles *via* neem extract and their anticancer and antibacterial activities

Hossam S. El-Beltagi[1,2], Marwa Ragab[3], Ali Osman[3], Ragab A. El-Masry[3], Khairiah Mubarak Alwutayd[4], Hind Althagafi[4], Leena S. Alqahtani[5], Reem S. Alazragi[5], Ahlam Saleh Alhajri[6] and Mahmoud M. El-Saber[7]

[1] Agricultural Biotechnology Department, College of Agriculture and Food Sciences, King Faisal University, Al-Ahsa, Saudi Arabia

[2] Biochemistry Department, Faculty of Agriculture, Cairo University, Giza, Egypt

[3] Biochemistry Department, Faculty of Agriculture, Zagazig University, Zagazig, Egypt

[4] Department of Biology, College of Science, Princess Nourah bint Abdulrahman University, Riyadh, Saudi Arabia

[5] Department of Biochemistry, College of Science, University of Jeddah, Jeddah, Saudi Arabia

[6] Food Science and Nutrition Department, College of Agricultural and Food Science, King Faisal University, Al-Ahsa, Saudi Arabia

[7] Biochemistry Unit, Genetic Resources Department, Desert Research Center, Cairo, Egypt

Corresponding authors
Hossam S. El-Beltagi,
helbeltagi@kfu.edu.sa
Ali Osman, aokhalil@zu.edu.eg

## ABSTRACT

In the present study, zinc oxide nanoparticles (ZnO-NPs) were synthesized using neem leaf aqueous extracts and characterized using transmission electron microscopy (TEM), ultraviolet visible spectroscopy (UV-Vis), and dynamic light scattering (DLS). Then compare its efficacy as anticancer and antibacterial agents with chemically synthesized ZnO-NPs and the neem leaf extract used for the green synthesis of ZnO-NPs. The TEM, UV-vis, and particle size confirmed that the developed ZnO-NPs are nanoscale. The chemically and greenly synthesized ZnO-NPs showed their optical absorbance at 328 nm and 380 nm, respectively, and were observed as spherical particles with a size of about 85 nm and 62.5 nm, respectively. HPLC and GC-MS were utilized to identify the bioactive components in the neem leaf aqueous extract employed for the eco-friendly production of ZnO-NPs. The HPLC analysis revealed that the aqueous extract of neem leaf contains 19 phenolic component fractions. The GC-MS analysis revealed the existence of 21 bioactive compounds. The antiproliferative effect of green ZnO-NPs was observed at different concentrations (31.25 μg/mL–1000 μg/mL) on Hct 116 and A 549 cancer cells, with an IC50 value of 111 μg/mL for A 549 and 118 μg/mL for Hct 116. On the other hand, the antibacterial activity against gram-positive and gram-negative bacteria was estimated. The antibacterial result showed that the MIC of green synthesized ZnO-NPs against gram-positive and gram-negative bacteria were 5, and 1 μg/mL. Hence, they could be utilized as effective antibacterial and antiproliferative agents.

## INTRODUCTION

Nanotechnology has become a promising technique in medicinal sciences, energy generation, nanoelectronics, and consumer products (*Rajendran & Mani, 2020*). Nanoparticle-based therapy is proposed as an alternate solution to address global challenges like antimicrobial resistance (AMR) and cancer (*Mariappan et al., 2021*). AMR is a significant worldwide public health issue, jeopardizing the efficacy of antibiotic treatment and creating obstacles to the development of new antibiotics. The incidence of morbidity and mortality resulting from drug-resistant diseases is increasing globally (*Serwecińska, 2020*). For this reason, the search for new antimicrobials from natural sources is crucial in modern medicine to combat the socioeconomic impact and health consequences of multidrug-resistant bacteria (*Mahgoub, Sitohy & Osman, 2013*; *El-Beltagi et al., 2019*; *Ebrahim et al., 2022*; *Fekry et al., 2022*). Consequently, the advancement of newer antimicrobial medications necessitates the adoption of a creative approach (*Abdel-Shafi et al., 2016*; *Abdel-Shafi et al., 2019a*; *Abdel-Shafi et al., 2019b*; *Abdel-Hamid et al., 2020b*; *Al-Mohammadi et al., 2020*; *Osman et al., 2021a*; *Abdel-Shafi et al., 2022*; *Enan et al., 2023*).

Another major disease and leading cause of death that has long been burdening the human population is cancer. Cancer is a prevalent disease caused by genetic abnormalities in cellular DNA, disrupting the process that controls cell death and division, resulting in uncontrollable cell growth in the human body (*Yadav & Mohite, 2020*). Traditional cancer therapies, including surgery, radiation, and chemotherapy, are successful but have significant side effects that diminish patients' quality of life (*Schirrmacher, 2019*; *Chandrasekaran, Anusuya & Anbazhagan, 2022*). Plant-based medicines can mitigate the harmful side effects of cancer treatment (*Abd Elhamid et al., 2022*).

Free radicals such as reactive oxygen species and/or reactive nitrogen species (ROS/RSN) generated during metabolic pathways cause oxidative stress (*Afify & El-Beltagi, 2011*; *Amer et al., 2022a*; *Amer et al., 2022b*). Oxidative stress is linked to chronic and degenerative diseases such as cancer, autoimmune and age-related disorders, cataracts, rheumatoid arthritis, cardiovascular and neurodegenerative diseases (*Abdel-Hamid et al., 2020a*; *Imbabi et al., 2021*; *Osman et al., 2021b*; *Reda et al., 2023*).

However, this process can be regulated by antioxidants produced naturally or externally supplied through foods and herbal supplements such as bioactive peptides (*Abdel-Rahim & El-Beltagi, 2010*; *Abdel-Hamid et al., 2017*; *Osman et al., 2019*; *El-Beltagi et al., 2020*).

Metal nanoparticles, including zinc oxide nanoparticles, have been effectively produced using environmentally friendly methods for a variety of uses, frequently demonstrating superior characteristics compared to those produced using traditional synthetic methods (*Schwartz, Marsh & Draelos, 2005*; *Iravani, 2011*). Zinc oxide nanoparticles (ZnO-NPs) are gaining significance in the healthcare sector due to their various benefits. Distinct antibacterial and wound healing properties, UV filtering capability, and strong catalytic and photochemical effects (*Nadhman et al., 2014*). The biosynthesis of ZnO-NPs by plants and fungi has been reported with antibacterial and antifungal properties (*Gunalan, Sivaraj & Rajendran, 2012*; *Najafi-Taher et al., 2018*). Several chemical and biological

techniques have been published for the synthesis of ZnO nanoparticles (NPs). In general, the most critical problem concerning the use of NPs in biomedical and biotechnological applications is the toxicity derived from the reducing agents and stabilizers used for the NP synthesis (*e.g.*, sodium borohydride, hydrazine, cetyltrimethylammonium bromide, various polymers *Hone et al., 2002*). To solve this problem, numerous researchers have concentrated on developing bio-friendly reagents for nanomaterial fabrication. Various natural compounds are being evaluated as nontoxic reducing agents or particle-surface stabilizers. Several natural compounds of plant origin (phytochemistry) are known to have anti-oxidant, anti-bacterial, and anti-inflammatory properties, as well as the ability to induce cell death in malignant cells (*Bellik et al., 2012*). However, green synthesis is one of the most environmentally friendly methods available (*Schwartz, Marsh & Draelos, 2005*; *Iravani, 2011*). Green synthesis of nanoparticles provides an advantage over other approaches since it is straightforward, one-step, cost-effective, environmentally friendly, and frequently results in more stable compounds (*Ahmed et al., 2016*).

*Azadirachta indica*, also known as neem, is highly valued in traditional Indian medicine due to its therapeutic benefits and rich phenolic content (*Kumar & Navaratnam, 2013*). *A. indica* has been used for centuries to treat a variety of ailments including inflammation, diarrhea, bacterial infections, and constipation. Its different parts, such as leaves, flowers, seeds, fruits, roots, and bark, have all been used for their therapeutic properties (*Duangjai et al., 2019*; *Ramadan et al., 2022a*; *Ramadan et al., 2022b*), cancer (*Paul, Prasad & Sah, 2011*), fever, and skin diseases (*Al Saiqali et al., 2018*). Additionally, *A. indica* has a wide range of pharmacological properties owing to its complex composition, which includes over 300 distinct bioactive chemicals with diverse activity (*Gupta et al., 2017*; *Abdel Rahman et al., 2023*; *Rahman et al., 2023*). The neem leaf extract, which contain functional substances such as cyclic peptides, sorbic acid, citric acid, phenolic compounds, polyhydroxy limonoids, ascorbic acid, retinoic acid, tannins, ellagic acid, and gallic acid, are thought to play an important role in the bioreduction and stabilisation of nanoparticles (*Narde et al., 2023*). Neem leaf extract contains phytochemicals such as flavones, organic acids, ketones, amides, and aldehydes. Flavones and organic acids, which are water-soluble, function as bio reductants and reduce zinc ions to form zinc nanoparticles (*Sangeetha, Rajeshwari & Venckatesh, 2011*).

Neem leaf extract contains phytochemicals such as flavones, organic acids, ketones, amides, and aldehydes. Flavones and organic acids, which are water-soluble, function as bio reductants and reduce zinc ions to form zinc nanoparticles (*Sangeetha, Rajeshwari & Venckatesh, 2011*). The main target of the present study was the green synthesis of ZnONPs *via* neem leaf aqueous extract and then comparing its efficacy as anticancer and antibacterial agents with chemically synthesized ZnONPs and the neem leaf extract used for the green synthesis of ZnONPs.

## MATERIALS & METHODS

### Plant materials

The leaves of *Azadirachta indica* A. Juss used in this study were provided by the Desert Research Centre (DRC) in Cairo, Egypt.

## Chemicals

DPPH (2,2-diphenyl-1-picrylhydrazyl; CAS Number: 84077-81-6), methanol; CAS Number: 67-56-1, zinc nitrate hexahydrate; CAS Number: 10196-18-6, sodium hydroxide; CAS Number: 1310-73-2, ethanol; CAS Number: 64-17-5, ethyl acetate; CAS Number: 141-78-6, gallic acid; CAS Number: 149-91-7, quercetin; CAS Number: 117-39-5, $AlCl_3$; CAS Number: 7446-70-0, Folin-Ciocâlteu reagent; Mfcd00132625, sodium carbonate; CAS Number: 497-19-8, and potassium acetate; CAS Number: 127-08-2 were acquired from Merck (KGaA, Darmstadt, Germany).

## Extract preparation

The neem leaves were gathered and washed thoroughly with distilled water and both surfaces of the leaves were sterilized using alcohol by gentle rubbing to avoid any contamination. The leaves were subsequently dehydrated for a period of three days under direct exposure to sunshine. The process involves pulverizing and filtering dried leaves using a mechanical blender (High-Capacity Upgraded Version ABS Mechanical Blender, 30,000 piece/pieces per month; Ningbo, China). The water extract of neem leaves is prepared by dissolving 10 gm of leaves in 100 mL of distilled water. The mixture is then heated on a hot plate magnetic stirrer (Benchmark H3770-HS Digital Hotplate Stirrer, temps up to 380 °C and speeds 150 to 1,500 rpm. Probe with direct feedback to microprocessor) to a temperature of 50 °C for 30 min. Afterward, the solution is filtered using filter paper (Whatman No. 1, pore size of 11 μm) and subjected to centrifugation (Sigma 2-6 Benxhtop Centrifuge) at a force of 5,000× g for 15 min. Immediately after that, zinc oxide NPs was prepared using aqueous extract of neem. The stock solution was stored in the refrigerator at 4 °C. This solution was also used for the investigation of neem phytochemistry (*Elumalai et al., 2012*; *El-Saber, 2021*; *Elshafie et al., 2023*).

## Chemically synthesis of ZnO NPs

Zinc oxide nanoparticles were synthesized using a chemical process using zinc nitrate and sodium hydroxide as starting materials, whereas sodium borohydride has been used as a stabilizing agent, as described by *Ahamed & Kumar (2016)*. In this experiment, a 0.1M solution of zinc nitrate (Zn $(NO_3)_2$.$6H_2O$) in water was stirred well continuously for one hour using a magnetic stirrer (Hotplate Magnetic Stirrer LHST-A11) to ensure complete dissolution of the zinc nitrate. Similarly, a 0.8M solution of NaOH in water was also stirred consistently for one hour. Once the zinc nitrate had fully dissolved, a 0.8M aqueous solution of NaOH was added gradually, drop by drop, over a period of 45 min manually, and then 1 ml NaBH4 (1%) was added dropwise after zinc nitrate dissolution. The solution was continuously stirred at a high speed during this process. The NaOH was completely added to the reaction, and then the reaction was allowed to proceed for aduration of 2 h. The beaker remained sealed in this condition for 4 h by aluminum foil at room temperature. Once the reaction was finished, the solution was left undisturbed overnight before the supernatant solution was meticulously separated. The remaining solution was subjected to centrifugation at 5,000× g for 10 min to eliminate the precipitate. The precipitate was then dried in an oven at 80 °C. $Zn(OH)_2$ was formed and then converted to ZnO.

The calcination had been performed at 500 °C for 1 h. The optical and nanostructured characteristics of the produced ZnO NPs nanoparticles were investigated.

## Green synthesis of ZnO NPs by neem leaves aqueous extract

The synthesis of ZnO-NPs was accomplished through the combination of 10 ml of neem leaf extract (as the reducing and capping agents) in water with 90 ml of a solution containing 1 mM zinc nitrate hexahydrate [Zn $(NO_3)_2$.6H$_2$O]. The mixture was heated on a hotplate magnetic stirrer at a temperature of 80 °C for 30 min, with continuous stirring. The existence of white-colored particles indicated the formation of nanoparticles. The particles obtained were subjected to centrifugation, and the resulting solid masses were gathered. The pellet was dried and calcined at 300 °C for 2 h. The process was followed to obtain white colored nanoparticles (*Pal et al., 2018*; *Zambri et al., 2019*; *El-Saber, 2021*; *Elshafie et al., 2023*).

## Green and chemically synthesized ZnO-NPs characterization
### *Transmission electron microscopy*

Actual morphology of the as-prepared ZnO-NPs was imaged by High-resolution transmission electron microscopy (HR-TEM) operating at an accelerating voltage of 200 kV (Tecnai G2; FEI). Diluted ZnO NPs suspension was ultra-sonicated for 5 min to reduce the particles aggregation. Using micropipette, about three drops from the ultra-sonicated solution were deposited on carbon coated-copper grid (200 mesh) and left to dry at room temperature. HR-TEM images of the ZnO NPs that were deposited on the grid were captured for morphological evaluation. All the preparation and characterization processes were conducted at the Nanotechnology and Advanced Materials Central Lab (NAMCL), Agricultural Research Center, Egypt.

### *Ultraviolet-Visible (UV-VIS) spectra*

The Shimadzu spectrophotometer, namely the UV-VIS module (UV-2450; Shimadzu), was employed to track the progression of ZnO-NPs formation inside the aqueous solution containing neem extract. The UV-Vis spectra were measured within the wavelength range of 300 to 700 nm.

## Dynamic Light Scattering (DLS), Zeta potential and particle size analysis

The average particle size distribution was determined using the DLS method, employing the zeta sizer (Malvern, ZS Nano, UK). The solution sample was placed in a cuvette (ZEN2112 Low-volume Quartz cuvette 20 μL) and treated until it became transparent, reducing the error in the reading. The device then measured the diameter of particles and determined the level of homogeneity using the PDI (dispersed index), which ranges from 0 to 1. A PDI approaching zero indicates increased homogeneity.

## Qualitative phytochemicals analysis of neem leaves aqueous extract

Following standard methods, preliminary qualitative phytochemical screening was performed (*Shrestha et al., 2015*).

### Alkaloids detection

The alkaloids were tested using Mayer's reagent (freshly prepared by dissolving a mixture of mercuric chloride (1.36 g) and potassium iodide (5.00 g) in 100 ml distilled water). A volume of two mL of botanical extract was introduced into a test tube, followed by the addition of 2–3 drops of Mayer's reagent. The detection of an alkaloid was indicated by the production of a green precipitate in the solution. Wagner's test was conducted utilizing Wagner's reagent (potassium iodide (2 g) and iodine (1.27 g) were dissolved in distilled water (5 mL) and the solution was diluted to 100 mL with distilled water). The presence of alkaloids was shown by the formation of a reddish-brown precipitate in a test tube holding 2 ml of extract upon addition of a few drops of Wagner's reagent.

### Flavonoids detection

A total of two mL of plant material was added to a test tube, along with an equal volume of a NaOH solution containing 2% w/v. A vivid yellow hue manifested within the test tube. The addition of a small amount of dilute hydrochloric acid resulted in the loss of colour, showing the existence of flavonoids. A volume of two mL of botanical extract was introduced into a test tube in preparation for the Shinoda Test. The substance underwent treatment with 5 drops of hydrochloric acid and 0.5 grammes of magnesium bits. The solution, which contained flavonoids, had a pink hue.

### Terpenoids detection

The obtained extract was dissolved in two mL of chloroform and subjected to evaporation until complete dryness. A volume of two mL of concentrated $H_2SO_4$ was introduced into the mixture. The presence of terpenoids is evidenced by the appearance of a reddish-brown colour at the boundary between the two substances.

### Saponins detection

Foam test was used to detect saponin. The concentrated solution was diluted with distilled water and transferred into a test tube. There was a temporary cessation for a few minutes. Saponins were detected within a foam layer of two cm in thickness.

### Steroids detection

The obtained extract was combined with chloroform (two mL), and then concentrated $H_2SO_4$ was added on the side. The manifestation of a crimson hue in the lower chloroform stratum signals the detection of steroids. Another experiment was carried out by combining the crude extract with two mL of chloroform. The mixture was then treated with 2 ml of concentrated $H_2SO_4$ andacetic acid. Steroid presence is signaled by the emergence of a green hue.

### Cardiac glycosides

Salkowski's test involved the combination of 2ml of crude extract with two mL of chloroform. The next step was a cautious addition of two mL of concentrated $H_2SO_4$, then gentle mixing. The presence of a reddish-brown color indicates the presence of a steroidal ring, specifically the aglycone portion of the glycoside.

The Keller-Kilani test involved mixing a crude extract with two mL of glacial acetic acid that contained 1–2 drops of a 2% $FeCl_3$ solution. The liquid was transferred into a separate test tube containing two mL of concentrated $H_2SO_4$. The presence of cardiac glycosides can be inferred from the existence of a brown ring at the contact.

## Polyphenols and tannins

The obtained extract was mixed with two mL of a 2% $FeCl_3$ solution. The blue–green or blue-black color suggested the presence of polyphenols and tannins.

## Total phenolic compounds determination

Total phenolic compounds (TPCs) were determined using the Folin–Ciocalteu method, according to the method of *Sánchez-Rangel et al. (2013)* with some modifications. The 0.3 mL from neem aqueous extract (500 µg/mL) was added and mixed with 1.2 mL of Folin and Ciocalteu's reagent (previously diluted 10-fold with distilled water). After 3 min, 1.5 mL of saturated $Na_2CO_3$ (75%) was subsequently added to the mixture. The samples were then incubated at 50 °C for 1 h, cooled and read at 765 nm. A calibration curve was created using a standard solution of gallic acid.

$$y = 0.001x + 0.0563 \ R^2 = 0.9792$$

where $y$ represents absorbance and $x$ represents gallic acid concentration in µg/mL.

All results were expressed as milligram of gallic acid per gram of extract (mg GAE/g extract).

## Total flavonoid content estimation

Total flavonoids content (TFC) in neem aqueous extract were analyzed with aluminum chloride method (*Ordoñez et al., 2006*) with a slight change. The assay mixture consisting of 0.5 mL of the neem aqueous extract (1,000 µg/mL), 0.5 mL distilled water, and 0.3 mL of 5% $NaNO_2$ was incubated for 5 min at 25 °C. This was followed by the addition of 0.3 mL of 10% $AlCl_3$ immediately. Two milliliters of 1 M NaOH was then added to the reaction mixture, and the absorbance was measured at 415 nm. The standard curve was created using quercetin. The total flavonoid levels were calculated using the calibration curve and the quercetin equivalent (QE).

$$y = 0.0012x + 0.008 \ R^2 = 0.944$$

where $x$ is the quercetin concentration in µg/mL and $y$ is the absorbance.

All results were expressed as milligram of quercetin per gram of extract (mg QE/g extract).

## Antioxidant activity estimation (DPPH-assay)

The antioxidant activity of the neem aqueous extract was assessed using the DPPH test method outlined by *Ramadan, Osman & El-Akad (2008)*. Briefly, 2.9 mL of a 0.1 mM DPPH methanolic solution was combined with each extract concentration (500, 1,000, 1,500, or 2,000 µg/mL). The reaction proceeded in dark for 30 min at 25 °C. The absorbance

of the combination was measured at 517 nm. The radical scavenging ability of DPPH was determined using the subsequent formula:

$$\text{Inhibition (\%)} = [(\text{Control Abs.} - \text{Sample Abs.})/\text{Control Abs.}] \times 100.$$

where Abs. control is the control absorbance and Abs. sample is the absorbance in the presence of extract.

## Phenolic compounds identification

High-performance liquid chromatography (HPLC) was used to identify polyphenolics. Compounds in the neem aqueous extract (*Abd Elhamid et al., 2022*). The phenolic compound detection was performed using an Agilent 1,260 Infinity HPLC system (Agilent, Santa Clara, CA, USA). The system was equipped with a quatpump (G 1311C), autosampler (G 1329B), column heater (G 1316A), variable wavelength detector (G 1314F), and an online degasser (G 1322A). The Agilent HPLC ChemStation 10.1 edition was utilized on a Windows 7 operating system to oversee instruments and conduct data analysis. The column used was an Agilent Zorbax C18 column with dimensions of 5 m in length and 4.6 mm in diameter, and a total length of 150 mm. The injection volume was 50 μL, and the Mobile Phase consisted of two components: A, which was a mixture of 70% methanol and 30% water, and B, which was 100% methanol. Samples were quantified by comparing the retention times with known authentic standards.

## Gas chromatography-mass spectrometry (GC-MS) analysis

Lyophilized neem crude extract (0.1 gm) was dissolved in 10 mL ethyl acetate. The neem ethyl acetate extract was examined *via* the GC-MS Agilent Technologies-7820A GC equipment. The capillary column used is the Agilent Technologies GC-MS HP-5MS, which has a length of 30 m, an inner diameter of 0.25 mL, and a film thickness of 0.25 m. The column is made of a 5% diphenyl and 95% dimethyl polysiloxane mixture. It is connected to the Agilent Technologies GCMS mass spectrometer model 5977MSD. A 70-eV electron ionization device was used. The carrier gas used was helium gas with a purity of 99.99%. The split ratio was 50:1, the injection volume was one mL, the injector temperature was set at 60 °C, and the ion source temperature was set at 250 °C. The temperature of the transfer line and ion source was set to 240 °C. The ionization mode used was electron impact at 70 eV. The scan time and scan interval were set to 0.2 s and 0.1 s, respectively.

The fragment size range is 40–600 Da. The components in the extracts were originally identified using peaks in the mass spectra utilizing Computer Wiley MS libraries, and these identifications were verified by comparing the two sets of data (*Balamurugan, Balakrishnan & Sundaresan, 2015*).

## Anticancer activity estimation
### Cell viability in vitro (MTT-assay)

The cell lines HCT116 (human colorectal cancer), and A549 (adenocarcinomic human alveolar basal epithelial cells), were obtained from Merck (KGaA, Darmstadt, Germany). The cells were sub-cultured in DMEM medium (Sigma-Aldrich) supplemented with 10% heat-inactivated foetal bovine serum (FBS), penicillin (10 U/mL), and streptomycin

(10 μg/mL). The cultures were kept in an incubator at 37 °C, 5% $CO_2$, and 100% humidity. $10 \times 10^3$ cells were put into each well of a 96-well microplate, and the cells grew for 24 h at 37 °C and 5% $CO_2$ before the samples were added. Different amounts (31.25–1000 μg/mL) of ZnO-NPs, neem extract, and green synthesised ZnO-NPs dissolved in distilled water were used to treat the cells. After 48 h of incubation, the absorbance at 550 nm was used to measure cell viability with the colorimetric MTT test (Promega, Madison, WI, USA) (*Hansen, Nielsen & Berg, 1989*). The positive control consisted of cells treated with a known volume of Triton X-100 (10 μL of a 10% solution), while the negative control consisted of cells left untreated. Percentages of cell viability and cytotoxicity were determined using the following equations:

Cell viability (%) = (Ab sample/Ab control) × 100

Cytotoxic activity(%) = 100% − cell viability(%).

The $IC_{50}$ value describes the concentration of a sample that results in a 50% reduction in growth.

## The quantification of caspase-9 mRNA expression

The Step-One Plus Real-time PCR, manufactured by Applied Biosystems in Foster City, CA, USA, was utilized to quantitatively analyze caspase-9 in two human cancer cell lines—A549 and HCT116. This analysis was carried out before and after treatments with ZnO-NPs, neem aqueous extract, and green synthesized ZnO-NPs, using gene-specific primers and SYBR Green master mix. The primers were designed using Oligo 7 software and tested for accuracy and specificity on the NCBI website. Cells were treated with different samples' $IC_{50}$ for 24 h. Quantitative real-time PCR was employed to assess the relative expression level of caspase 9. The average score of duplicated Ct values was measured for each sample, and the relative expression levels of the target genes were determined using the comparative Ct method. The caspase 9-forward and caspase 9-reverse primer sequences were 5′-GCAGGCTCTGGATCTCGGC-3′ and 5′-GCTGCTTGCCTGTTAGTTCGC-3′, respectively. The annealing temperatures for these primers were 60.5 and 59.5 °C, respectively (*Asadi et al., 2018*).

## Antibacterial activity
### Minimum inhibitory concentration estimation

The minimum inhibitory concentration (MIC) of neem aqueous extract, ZnO-NPs, and green synthesized ZnO-NPs against gram-positive (*Listeria monocytogens* and *Staphylococcus aureus*) and gram-negative (*Salmonella Enteritidis* and *Escherichia coli*) bacteria was determined using the microdilution method. To experiment, 20 μL of 24-hour-old bacterial culture from well number 1 was added to each well of a 96-well plate. Following that, 100 μL of the tested samples were put into each well along with the same amount of Mueller-Hinton broth (MHB). The concentrations were 20, 10, 5, 2, 1, and 0.5 μg/mL. To serve as a control, a solution devoid of particles was utilized. The plates

were left to incubate for the duration of the night. A 4% w/v p-iodonitro-tetrazolium violet solution (INT, Sigma-Aldrich) was added to 20 μL of each well to show that bacteria were present. The MIC was defined as the lowest concentration of the sample that effectively inhibited the development of microorganisms (*Tayel et al., 2010*).

### Transmission electron microscopy

*Staphylococcus aureus* and *Escherichia coli* were cultivated in Mueller Hinton Broth (MHB) and incubated at 37 °C until they reached a maximum concentration of $10^9$ CFU mL$^{-1}$. The culture was then diluted to $10^8$ CFU mL$^{-1}$ using a peptone solution (0.1% with 0.85% NaCl). The neem aqueous extract, ZnO-NPs, and green-synthesised ZnO-NPs were added to the cell suspensions, except for the control, and incubated at 37 °C for 4 h. Bacterial cells were collected, fixed, washed, and dehydrated before examination using a transmission electron microscope (JEOL-TME-2100F) as described by *Sitohy et al. (2013)*.

## Statistical analysis

The recorded data was analyzed using version 21 of the Statistical Package for Social Sciences (SPSS), developed by SPSS Inc. (Chicago, IL, USA). The analysis was conducted following the guidelines provided (*Dytham, 2011*). The parameters were subjected to a one-way analysis of variance (ANOVA) test and the Duncan test to assess the statistical significance of the differences in means. Data were deemed statistically significant if the $P \leq 0.05$.

## RESULTS

### Characterization of ZnO-NPs prepared chemically and green synthesis

#### Characterization of ZnO-NPs prepared chemically

The analysis of the generated ZnO-NPs showed that they were disseminated in an essentially spherical shape with a particle size of approximately 85 ± 5 nm, as observed using transmission electron microscopy (TEM), as shown in Fig. 1A. UV-VIS optical absorption of ZnO-NPs. Figure 1B displays the UV-VIS absorption spectrum of ZnO-NPs. The absorbance spectra of the colloidal ZnO NPs solution displays a surface Plasmon resonance peak at 328 nm, indicating successful production and dispersion of the nanoparticles in the aqueous solution without any aggregation. In Fig. 1C, DLS was used for additional confirmation to assess the uniformity and estimate the average particle size, which was found to be 85 ± 5 nm. The polydisperse index (pdi) of 0.552 indicates a limited range of sizes. Based on Fig. 1D, the Zeta potential is measured to be −11.5 mV, indicating that the NPs are somewhat stable in terms of their stability and surface charge.

#### Characterization of ZnO-NPs green synthesis by aqueous neem leaves extract

With increasing incubation time, neem leaf extracts biosynthesized ZnO-NPs, resulting in a color change from translucent to yellowish brown, indicating bio reduction of zinc oxide particles. The color shift is caused by the stimulation of surface plasmon resonance in solution. Figure 2A shows the surface morphologies and particle sizes of ZnO synthesized

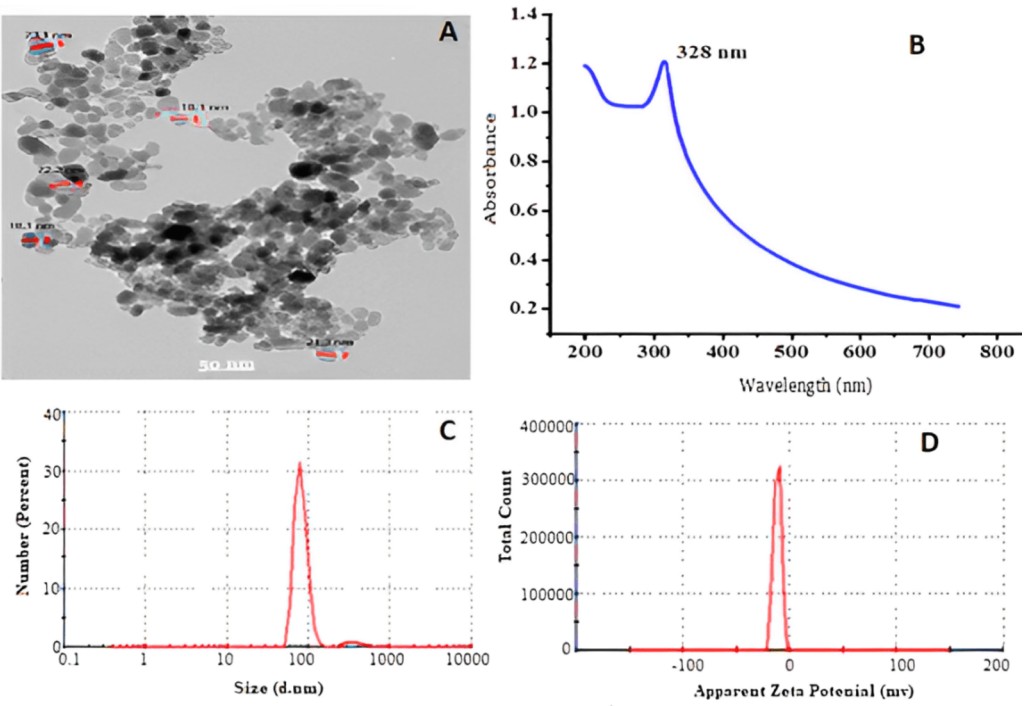

**Figure 1** Transmission electron microscope (A), absorbance of ZnO-NPs UV-VS (B) particle size (C), and zeta potential (D) of ZnO NPs prepared using chemical method.

using neem extract by TEM. The influence of the preparation process on the photo-optical properties of zinc oxide nanoparticles was investigated using their ability to absorb UV radiation. Figure 2B depicts the UV-visible absorption spectra of zinc oxide nanoparticles. UV-visible spectral investigation yielded absorption spectra of 380 nm. In Fig. 2C, the particle size of synthesized ZnO powder is approximately 62.5 nm, and the zeta potential is shown in Fig. 2D.

### Qualitative phytochemical screening of neem leaf aqueous extract

*Azadirachta indica* has gained significance in the contemporary world setting due to its ability to provide solutions to the significant challenges confronting humanity. It is a fast-growing evergreen popular tree found commonly in India, Africa, and America. As presented in Table 1, qualitative analysis for leaf aqueous extract of *A. indica* shows the presence of terpenoids, saponins, and steroids having the highest concentration, while cardiac glycosides and tannins have moderate concentrations however alkaloids, flavonoids, and phenols have low concentrations. Therapeutic plants include a variety of secondary metabolites such as alkaloids, flavonoids, saponins, and other active compounds. These compounds have significant therapeutic properties and have been widely utilised in the medicine and pharmaceutical sectors. These secondary metabolites are documented to possess numerous biological and medicinal characteristics.

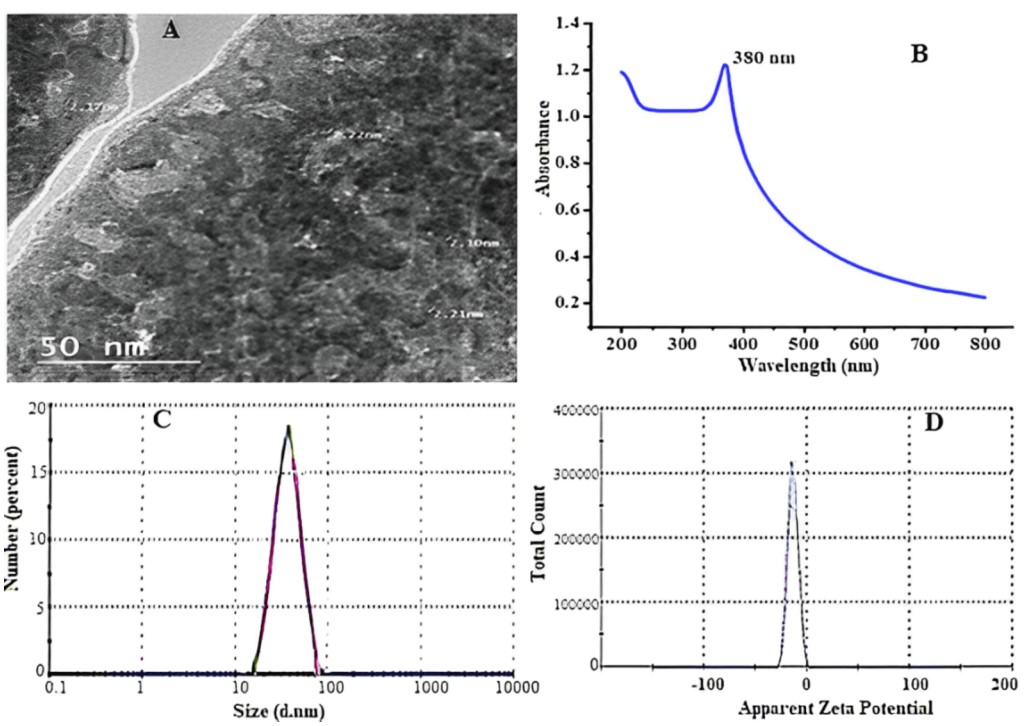

**Figure 2** Transmission electron microscope (A), absorbance of ZnO NPs UV-VIS (B) particle size (C), and zeta potential (D) of ZnO-NPs prepared using green synthesis.

**Table 1** Qualitative phytochemical screening of neem leaf aqueous extract.

| No. | Components | Abundance |
|---|---|---|
| 1 | Alkaloids | + |
| 2 | Flavonoids | + |
| 3 | Terpenoids | +++ |
| 4 | Saponins | +++ |
| 5 | Steroids | +++ |
| 6 | Cardiac Glycosides | ++ |
| 7 | Phenols | + |
| 8 | Tannins | ++ |

**Notes.**
While: (+++) Most present; (++) Moderately present; (+) Least present.

## Total phenolic (TPC), total flavonoids (TFC) content, and antioxidant activity

Data presented in Table 2 show that the quantitative of total phenolic (TPCs by mg GAE $g^{-1}$ dry extract), total flavonoid (TFCs by mg QE $g^{-1}$ dry extract), and antioxidant activity (DPPH %) of neem leaf aqueous extract at different concentrations (500–1,000–1,500–2,000 μg/mL). Data declared that significant differences ($P \leq 0.05$) were examined between the phytochemical content of *A. indica* leaves. The mean value for TPC in *A. indica* leaves was calculated as 119.00 mg GAE $g^{-1}$ dry extract. The TFC content of *A. indica* leaves

**Table 2** Quantitative total phenolic, total flavonoid and antioxidant activity (DPPH %) of neem leaf aqueous extract at different concentrations (500–1,000–1,500–2,000 μg/mL).

| Parameters | Neem variants |
|---|---|
| TPCs (mg GAE g$^{-1}$ dry) | 119.00 ± 1.3[a] |
| TFs (mg QE g$^{-1}$ dry extract) | 45.67 ± 0.95[b] |
| Antioxidant activity (%) | % Inhibition |
| 500 | 31.67 ± 0.25[d] |
| 1,000 | 46.76 ± 0.87[c] |
| 1,500 | 65.33 ± 1.09[b] |
| 2,000 | 77.83 ± 2.11[a] |

Notes.
Different letter in the same row indicates significant difference ($p > 0.05$).
mg GAE g-1, mg gallic acid equivalents per g dry weight; mg EQ g-1, mg Quercetin equivalents per g dry weight; % Inhibition, Inhibition of DPPH radical.

was observed as 45.67 mg QE g$^{-1}$ dry extract. To complete the data for the aqueous leaves of *A. indica*, it is necessary to measure their antioxidant activity using DPPH. The percentage of antioxidant activity of *A. indica* leaf extract was measured using DPPH assay at various concentrations, including 500, 1,000, 1,500, and 2,000 μg/mL, and results are presented in Table 2. As the concentration of *A. indica* leaf extract increased, its antioxidant activity demonstrated a gradual increase. As the extract concentration increased from 500 to 2,000 μg/mL, the effectiveness of DPPH radical scavenging increased from 31.67% to 77.83%. The previous results reaffirmed our viewpoint regarding the antioxidant properties of plant extracts, which are attributed to the presence of polyphenolic compounds. These compounds have potential as effective antioxidant agents. Moreover, it is widely recognized that plant extracts containing phenolic and flavonoid components exhibit a substantial number of antioxidants. Phenolic compounds or polyphenols are produced by plants because of their secondary metabolism. These chemicals are frequently present in plants and have been extensively utilized due to their diverse biological actions, which include antioxidant properties.

## Phenolic compounds content in neem extract by HPLC

The quantity of phenolic compounds in neem aqueous extract was found in (Table 3). The concentrations of the components that were found varied greatly. According to the standards used in the HPLC, the aqueous extract of A. indica leaves contained 19 phenolic compound fractions, which included gallic acid, chlorogenic acid, catechin, methyl gallate, coffeic acid, syringic acid, pyrocatechol, rutin, ellagic acid, coumaric acid, vanillin, ferulic acid, naringenin, daidzein, quercetin, cinnamic acid, apigenin, kaempferol, and hesperetin. In the connect, chlorogenic acid (896.76 μg/g), catechin (666.25 μg/g), gallic acid (456.68 μg/g), vanillin (423.68 μg/g), and hesperetin (372.15 μg/g) are the key five components. In addition, coumaric acid (9.75 μg/g) and cinnamic acid (8.80 μg/g) had the lowest value, while pyrocatechol and kaempferol did not detect. This variation in phenolic compounds may be responsible for this extract's greater antioxidant effect in reducing oxidation. As a result, phenolic acids can protect against a wide range of oxidative damaged diseases.

**Table 3  Phenolic compounds content (μg/g extract) in neem leaf aqueous extract.**

| N0 | RT | Phenolic compound | Conc. (μg/g ) |
|---|---|---|---|
| 1 | 3.381 | Gallic acid | 456.68 |
| 2 | 4.329 | Chlorogenic acid | 896.76 |
| 3 | 4.671 | Catechin | 666.25 |
| 4 | 5.58 | Methyl gallate | 103.65 |
| 5 | 5.868 | Coffeic acid | 38.23 |
| 6 | 6.496 | Syringic acid | 52.06 |
| 7 | 6.787 | Pyro catechol | 0.00 |
| 8 | 7.954 | Rutin | 149.84 |
| 9 | 8.562 | Ellagic acid | 52.80 |
| 10 | 9.395 | Coumaric acid | 9.75 |
| 11 | 10.13 | Vanillin | 423.68 |
| 12 | 10.301 | Ferulic acid | 189.56 |
| 13 | 10.535 | Naringenin | 165.28 |
| 14 | 11.909 | Daidzein | 28.12 |
| 15 | 12.754 | Querectin | 41.16 |
| 16 | 14.082 | Cinnamic acid | 8.80 |
| 17 | 14.515 | Apigenin | 135.75 |
| 18 | 15.032 | Kaempferol | 0.00 |
| 19 | 15.673 | Hesperetin | 372.15 |

## GC-MS analysis of neem leaves ethyl acetate extract

The GC-MS analysis of neem leaf extract ethyl acetate fraction revealed 21 peaks, indicating the presence of 21 bioactive compounds. The chromatogram is presented in (Fig. 3, and Table 4) lists the bioactive compounds together with their retention time (RT), peak areas (%), molecular formula, and molecular weight (MW). The bioactive compounds revealed are $\alpha$-terpinolene; citronellyl propionate; hexadecanoic acid, ethyl ester; palmitic acid, TMS derivative; phytol; distearyl phosphate; $\alpha$-linolenic acid, TMS derivative; dicyclohexyl phthalate; heptacosane; tetracosane; pent-4-enal; Vitamin E; ethyl 2-cyano-3-Ethylpentanoate; stigmasterol; $\gamma$-sitosterol; $\beta$-sitosterol, TMS derivative; 2-diazocyclooctanone; 3-butenamide, 4-(4-chlorophenyl)-N-(1,1-dimethylethyl)-3-methyl-4-phenyl-, (Z)-; andrographolide; (2S,3R)-2,3-epoxy-5-methyl-5-hexene-1-ol and stigmasteryl tosylate. Among the twenty-one compounds identified, the major compounds present in neem ethyl acetate extract were distearyl phosphate, andrographolide, phytol, and (2S,3R)-2,3-epoxy-5-methyl-5-hexene-1-ol.

## Cytotoxicity on A549 and HCT116

To explore the possible cytotoxic effects of neem aqueous extract, ZnO-NPs, and green synthesized ZnO-NPs on colorectal (HCT116) and lung (A549) cancer cells, different concentrations of the extract and nanoparticles were used to treat the cells, and the resulting antiproliferative effects were measured using MTT assay. Following the administration of neem aqueous extract, ZnO-NPs, and green synthesized ZnO-NPs, the morphology

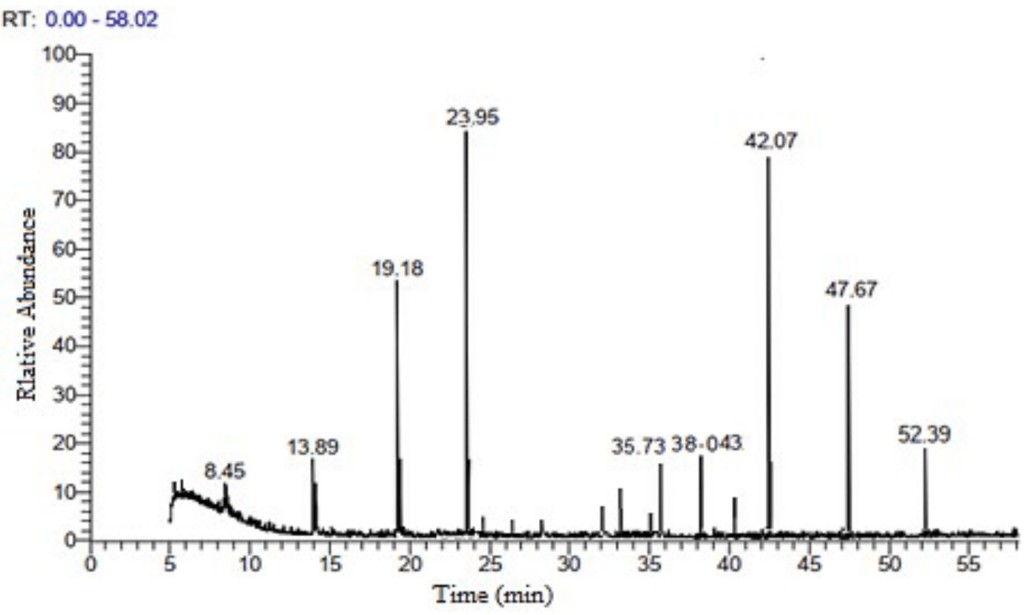

**Figure 3  GC-MS chromatogram of neem leaves ethyl acetate extract.**

of A549 and HCT116 cell lines exhibited aberrations in comparison to the untreated cell control. The application of neem aqueous extract, ZnO-NPs, and green-synthesized ZnO-NPs resulted in observable morphological alterations in the treated cells, including cell shrinkage and suspension in the media (Figs. 4 and 5).

## MTT-assay

The toxicity (%) and cell viability (%) of the A549 and HCT116 cell lines, when subjected to treatment with ZnO-NPs, neem aqueous extract, and green synthesized ZnO-NPs at different concentrations, are depicted in Figs. 6A and 6B. The linear correlation between cell viability and toxicity from the tested substances is presented in Figs. 4 & 5. The data clearly demonstrates a negative correlation between the concentration of the tested materials and the overall cell viability percentage. The MTT assay demonstrated that the tested materials exhibited a concentration-dependent inhibitory effect on the growth of human cancer cell lines (A549 and HCT116). The findings demonstrate that both the extract and nanoparticles exhibited a dose-dependent inhibition of cell proliferation, as illustrated in Figs. 4, 5, 6A, 6B and 7. Our study revealed that the antiproliferative effect of green synthesized Zn-NPs was much superior to that of ZnO-NPs and neem aqueous extract. This suggests that the cytotoxic activities of neem aqueous extract and ZnO-NPs were enhanced upon their conjugation. Figures 6A and 6B demonstrate that ZnO-NPs can reduce the cell viability percentage of A549 and HCT16 cell lines.

The 50% inhibitory concentration ($IC_{50}$) values of ZnO-NPs, neem extract, and green synthesized ZnO-NPs for A549 and HCT116 cell lines after 24 h of treatment are presented in Fig. 7. The neem aqueous extract had the lowest $IC_{50}$ against A 549 (227 µg/mL), with

**Table 4  GC-MS analysis of neem (*Azadirachta indica* A. Juss) leaf ethyl acetate extract.**

| PN | RT | Active compounds | MF | MW | Area % |
|---|---|---|---|---|---|
| 1 | 5.18 | α-Terpinolene | $C_{10}H_{16}$ | 136.23 | 0.38 |
| 2 | 5.22 | Citronellyl propionate | $C_{13}H_{24}O_2$ | 212.33 | 0.78 |
| 3 | 8.45 | Hexadecanoic acid, ethyl ester | $C_{18}H_{36}O_2$ | 284.47 | 3.12 |
| 4 | 13.89 | Palmitic Acid, TMS derivative | $C_{19}H_{40}O_2Si$ | 328.6 | 3.7 |
| 5 | 19.18 | Phytol | $C_{20}H_{40}O$ | 296.53 | 9.46 |
| 6 | 23.95 | Distearyl phosphate | $C_{36}H_{75}O_4P$ | 603 | 29.81 |
| 7 | 24.11 | α-Linolenic acid, TMS derivative | $C_{21}H_{38}O_2Si$ | 350.61 | 1.00 |
| 8 | 26.31 | Dicyclohexyl phthalate | $C_{20}H_{26}O_4$ | 330.42 | 0.42 |
| 9 | 28.25 | Heptacosane | $C_{27}H_{56}$ | 380.7 | 0.63 |
| 10 | 28.38 | Tetracosane | $C_{24}H_{50}$ | 338.65 | 2.08 |
| 11 | 32.16 | Pent-4-enal | $C_5H_8O$ | 84.12 | 0.96 |
| 12 | 33.21 | Vitamin E | $C_{29}H_{50}O_2$ | 430.71 | 2.72 |
| 13 | 35.36 | Ethyl 2-Cyano-3-Ethylpentanoate | $C_{10}H_{17}NO_2$ | 183.25 | 0.8 |
| 14 | 35.73 | Stigmasterol | $C_{29}H_{48}O$ | 412.7 | 3.65 |
| 15 | 38.043 | γ-Sitosterol | $C_{29}H_{50}O$ | 414.7 | 4.18 |
| 16 | 38.45 | β-Sitosterol, TMS derivative | $C_{32}H_{58}OSi$ | 486.88 | 1.9 |
| 17 | 39.02 | 2-Diazocyclooctanone | $C_8H_{12}N_2O$ | 152.19 | 1.96 |
| 18 | 40.76 | 3-Butenamide, 4-(4-chlorophenyl)-N-(1,1-dimethylethyl)-3-methyl-4-phenyl-, (Z)- | $C_{21}H_{24}ClNO$ | 341.88 | 0.89 |
| 19 | 42.07 | Andrographolide | $C_{20}H_{30}O_5$ | 350.4 | 18.82 |
| 20 | 47.67 | (2S,3R)-2,3-Epoxy-5-methyl-5-hexene-1-ol | $C_7H_{12}O_2$ | 128.17 | 8.53 |
| 21 | 52.39 | Stigmasteryl tosylate | $C_{36}H_{54}O_3S$ | 566.9 | 4.22 |
| | | | Total area percent | | 100.01 |

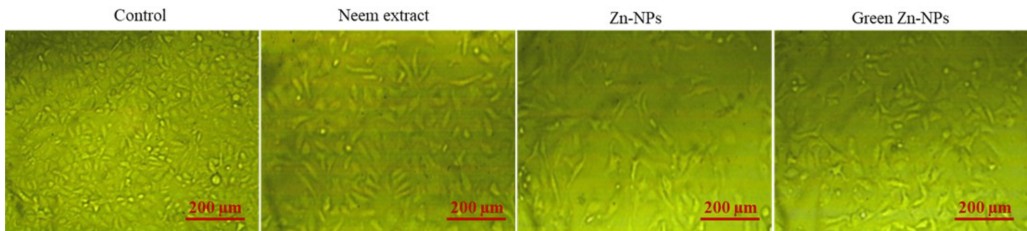

**Figure 4  The shape of HCT 116 cells changed after being treated for 24 h with neem aqueous extract, Zn-NPs, and green synthesized ZnO-NPs at IC$_{50}$ concentrations (350, 190, and 118 μg/mL, respectively).**

HCT16 coming in second (350 μg/mL). On the other hand, Zn-NPs had the lowest IC$_{50}$ against HCT116 (190 μg/mL), with A 549 coming in second (197 μg/mL). Finally, the IC$_{50}$ value for green Zn-NPs was found to be 111 μg/mL for A 549 and 118 μg/mL for HCT116.

The utilization of green synthesis techniques in the production of ZnO-NPs has been found to have a notable impact on the upregulation of caspase-9 transcript expression (Fig. 8). This effect is observed to be much greater when compared to other treatments involving neem extract and Zn-NPs, as well as the control group consisting of untreated

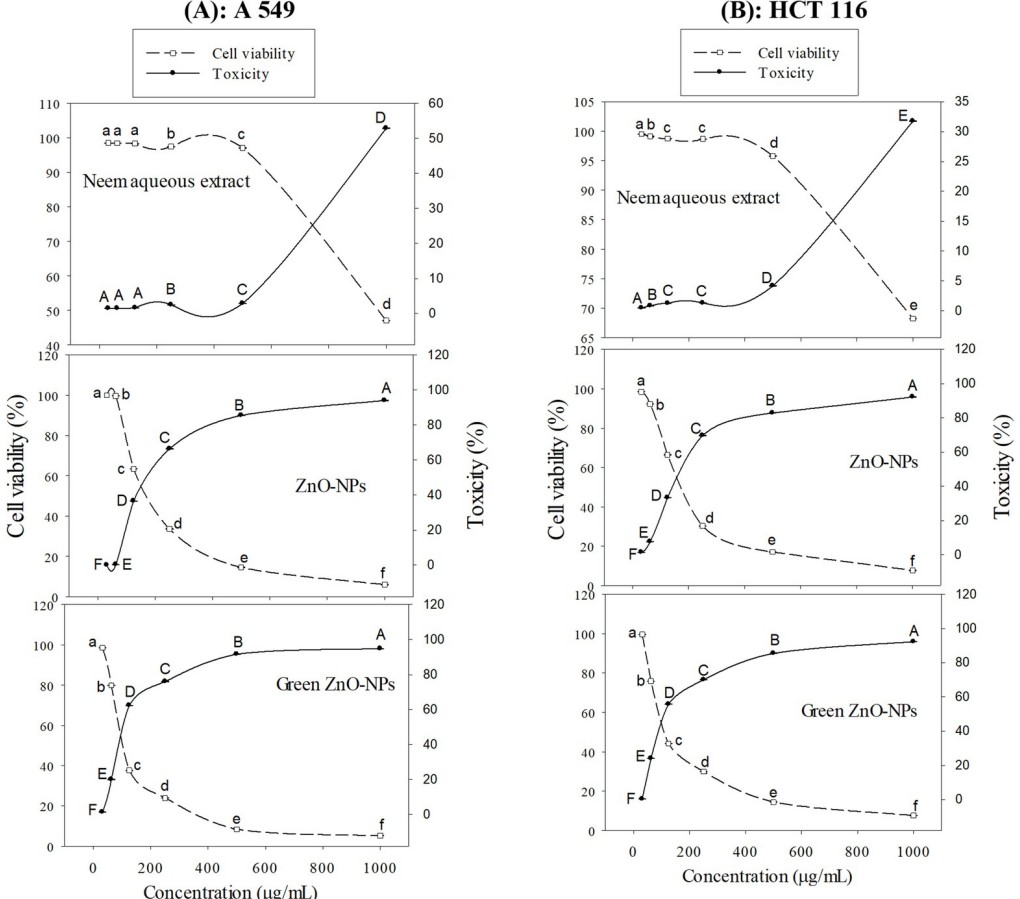

**Figure 5** The shape of A549 cells changed after being treated for 24 h with neem aqueous extract, Zn-NPs, and green synthesized ZnO-NPs at $IC_{50}$ concentrations (227, 197, and 111 μg/mL, respectively).

**Figure 6** Cell viability (%) and Toxicity (%) of A549 (A), and HCT 116 (B) cell lines treated with ZnO-NPs, neem aqueous extract, and green synthesized ZnO-NPs at different concentrations. Different letters indicate significant differences among the cell viability (small letters) and toxicity (capital letters) according to Tukey's HSD test ($p \leq 0.05$).

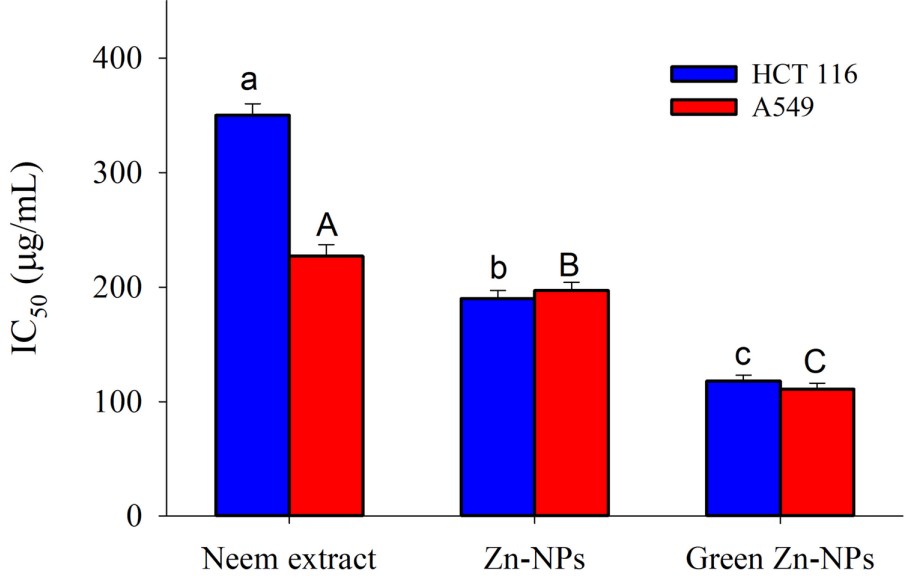

**Figure 7** **The 50% inhibitory concentration (IC$_{50}$) values of ZnO-NPs, neem extract, and green synthesized ZnO-NPs for A549 and HCT 116 cell lines after 24 hours of treatment.** The different letters indicate significant differences among the IC50 against HCT 116 (small letters) and the IC50 against A549 (capital letters), according to Tukey's HSD test ($p \leq 0.05$).

cells. Treating A549 and HCT116 cells with 111 and 118 µg/mL Zn-NPs increased caspase-9 expression by 7.14 and 7.53-fold, respectively, after 24 h compared to untreated controls.

## Antibacterial activity
### MIC estimation
Different concentrations of neem extract, ZnO-NPs, and green synthesized ZnO-NPs (0, 0.5, 1, 2, 5, 10, and 20 µg/mL) were tested for antibacterial activity. The results showed that the MIC of neem aqueous extract, ZnO-NPs, and green synthesized ZnO-NPs against Gram-positive *S. aureus* and *L. monocytogenes* were 10, 5, and 1 µg/mL, respectively (Table 5). The MIC against Gram-negative bacteria *S. Enteritidis* and *E. coli* was 20, 10, and 5 µg/mL, respectively (Table 5).

### TEM
Following the addition of neem aqueous extract, ZnO-NPs, and green-synthesized ZnO-NPs (at a concentration of 1 MIC) to MHB media containing *Staph. aureus* and *E. coli*, TEM pictures revealed a reduction in the proportion of intact cells after 4 h of incubation at 37 °C (Fig. 9).

The TEM pictures showed that bacterial cells that had survived displayed diverse deformations. The compounds subjected to testing (1 MIC) exhibited comparable impacts on *Staph. aureus*, including cellular contraction, wrinkling of the cell membrane, creation of pores, and depletion of viable cellular matter. Comparable findings were likewise noted for *E. coli*.

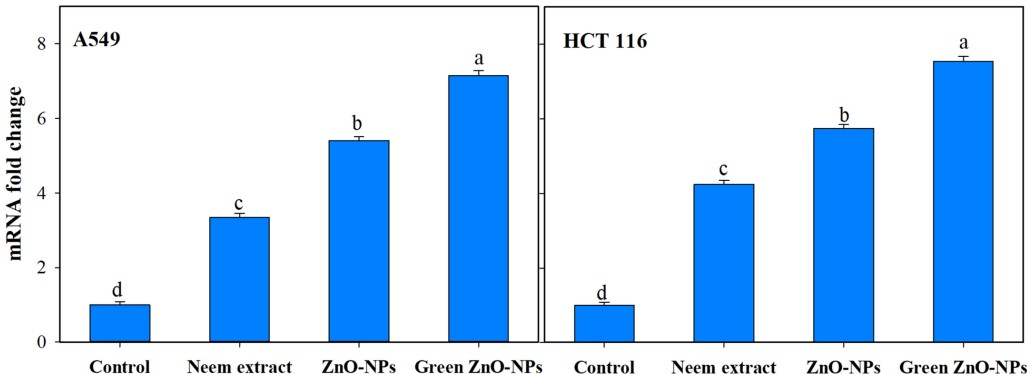

**Figure 8** **Effect of neem aqueous extract, ZnO-NPs, and green synthesized ZnO-NPs on caspase 9 gene expression of two human cancer cell lines (A549, and HCT 116).** Cells were treated with the concentration causing the IC$_{50}$ for each cell line for 24 h and their mRNA levels were evaluated by quantitative real-time PCR. The different letters indicate significant differences among treatments, according to Tukey's HSD test ($p \leq 0.05$).

**Table 5** **Minimum inhibitory concentration (MIC) of neem aqueous extract, ZnO-NPs, and green synthesized ZnO-NPs against gram positive (*L. monocytogenes* and *Staph. Aureus*) and gram negative (*S. Enteritidis* and *E. coli*) bacteria.**

| Bacterial strains | MIC (μg/mL) | | |
|---|---|---|---|
| | **Neem extract** | **ZnO-NPs** | **Green ZnO-NPs** |
| | Gram positive | | |
| *S. aureus* | 10 | 5 | 1 |
| *L. monocytogenes* | 10 | 5 | 1 |
| | Gram negative | | |
| *E. coli* | 20 | 10 | 5 |
| *S. Enteritidis* | 20 | 10 | 5 |

# DISCUSSION

Nanoparticles can be synthesized using a variety of methods; nonetheless, environmentally friendly methods have become chosen over traditional chemical and physical procedures. The synthesis of nanoparticles by chemical and physical methods can be expensive, time-consuming, and energy-intensive, have a detrimental influence on the environment, and leave toxic compounds on the surface. These nanoparticles cannot be employed in medicinal applications (*El-Belely et al., 2021*). This study used a biological technique to synthesize ZnO-NPs. *Azadirachta indica* leaves aqueous extract was utilized as a reducing and stabilizing agent to synthesize ZnO-NPs. Plant-based synthesis methods have several benefits, such as being easy to deal with, affordable, and possible without the use of chemical solvents or harmful chemicals (*Sohail et al., 2020*; *Fouda, 2023*). The phytochemical analysis of neem extract indicates that it consists of several distinct components (*Gupta et al., 2017*). This plant group is particularly rich in phenolic chemicals, which are employed as a reducing agent in the synthesis of ZnO-NPs. Although the exact mechanism for forming ZnO-NPs

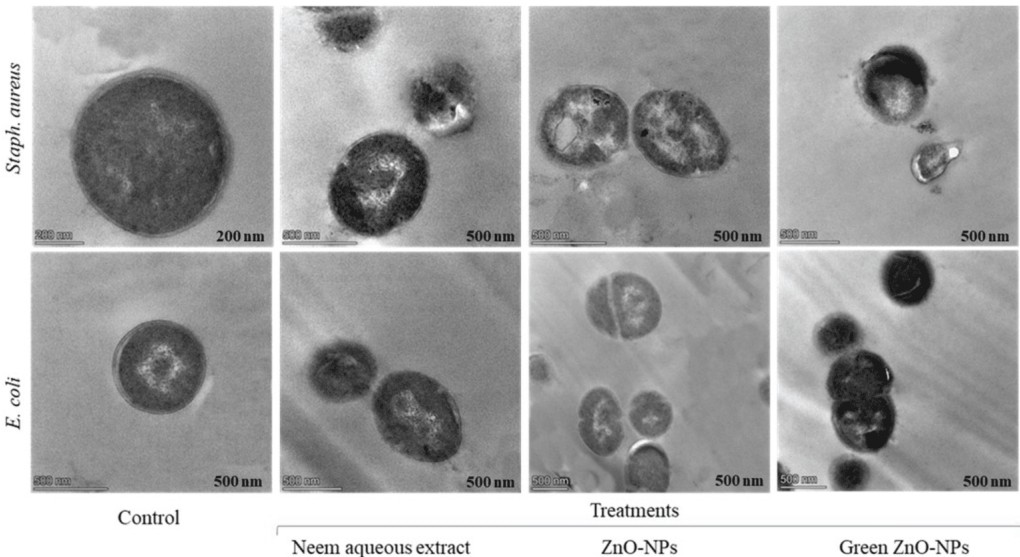

**Figure 9 Transmission electron microscopy graphs *Staph. aureus*, and *E. coli* as subjected to 1 MIC of neem aqueous extract, ZnO-NPs, and green synthesized ZnO-NPs.**

from plant extracts is unknown, it is thought that polar groups play a role (*Jha & Prasad, 2010*; *Anandan et al., 2019*). Hence, one of the plausible mechanisms for the reducing and capping effects of the plant extract during the formation of ZnO-NPs is represented in Fig. 10. According to the reduction of the zinc nitrate using *Azadirachta indica* extract as a green synthesis of ZnO-NPs, the compounds containing (–OH, -NH, and $NH_2$) can reduce ($Zn^{2+}$) ion. They exhibit appropriate reducing effects in addition to high stabilizing properties during ZnONPs preparation (*Anandan et al., 2019*; *Alharbi, Abaker & Makawi, 2023*; *Eltak et al., 2023*). The chemical and green synthesis of ZnO-NPs has been extensively explored and confirmed using TEM, UV-Vis spectrophotometer, and DLS analysis. . These results are similar to some results of studies that used neem extract to prepare zinc oxide nanoparticles, but with different sizes. In this regard, the sizes formed of ZnO-NPs were 9.6 to 25.5 nm (*Bhuyan et al., 2015*), and 19.57 ± 1.56 nm (*Sohail et al., 2020*).

The analysis of phytochemicals in the neem aqueous extract: phenolics, flavonoids, alkaloids, tannins, terpenoids, saponins, steroids, and cardiac glycosides were similar to earlier findings (*Dash, Dixit & Sahoo, 2017*). Plant secondary metabolites, known as phytoconstituents, are a useful and unique source of food supplements and medications. Extensive research has thoroughly established their numerous functions and uses (*García-Cruz, Salinas-Moreno & Valle-Guadarrama, 2012*). In the present study, some of the main phenolic and flavonoids found in the extract by HPLC analysis are hesperetin, naringenin, cinnamic acid, apigenin, kaempferol, naringenin, coumaric acid, vanillin, ferulic acid, naringenin, daidzein, quercetin, cinnamic acid, apigenin, kaempferol, and apigenin. These compounds have antioxidant (*Sani & Baburo, 2020*; *Nagano & Batalini, 2021*), antibacterial (*Elizabeth Babatunde et al., 2019*), and anticancer (*Azhagu Madhavan, 2021*; *Madhavan, 2021*) properties. The inquiry used GC-MS to analyze the ethyl acetate

**Figure 10** The plausible mechanism of formation of zinc oxide nanoparticles (ZnO-NPs) from neem aqueous leaf extract.

fraction of neem leaf extract. The research findings revealed the presence of 21 bioactive compounds. Although 21 compounds occurred, only 14 had a high area percent (peaks), with the remaining 7 having a low area percent, as seen in Fig. 3. The neem ethyl acetate extract contains hydrocarbons, terpenoids, phenolics, alkaloids, fatty acids, and their derivatives. The literature study indicates that most of the prevalent compounds in neem have biological activity (*Cock et al., 2009*; *Lucantoni et al., 2010*; *Hossain & Nagooru, 2011*). Phenols and flavonoids have an aromatic ring with at least one hydroxyl group substituted. It forms chelate compounds with metal ions. Consequently, they are prone to oxidation. Therefore, they serve as great entities for donating electrons (*Sankhalkar, 2014*).

To assess the anticancer activity of green and chemically synthesized ZnONPs compared to neem leaf aqueous extract, the MTT assay was used. Based on the results of this study, the eco-friendly ZnO-NPs had the biggest effect, as shown by the lowest $IC_{50}$ value. They were followed by ZnO-NPs and then the neem aqueous extract. The improved surface chemistry of ZnO-NPs synthesized using green methods and ZnO-NPs, when compared to neem aqueous extract, can be linked to their reduced size. The variation in size may enhance their contact with cancer cells (*Mthana et al., 2022*). While numerous research has documented the anticancer properties of ZnO-NPs, the precise underlying mechanism remains to be fully understood. ZnO-NPs have been shown to help fight cancer. This is likely because they create reactive oxygen species (ROS) on their surface. Furthermore, the breaking down

of the particles and the subsequent release of free $Zn^{2+}$ ions help cells make ROS (*Kasemets et al., 2009*; *Wang et al., 2014*; *El-Beltagi et al., 2022*). Several studies have shown that ZnO NPs can kill cells by creating ROS (*Sirelkhatim et al., 2015a*). There is evidence from several studies that ROS cause lipid peroxidation, enzyme inactivation, and membrane degradation. The main reasons why ZnO NPs are thought to fight cancer are these mechanisms (*Yu et al., 2013*; *Wang et al., 2015*). To find alternatives that are better for the environment, scientists are looking into long-term ways to make metal nanoparticles and metal oxide nanoparticles using plant extracts. The approaches are characterized by their relative ease, cost-effectiveness, and environmental superiority as compared to chemical and physical alternatives (*Alshameri & Owais, 2022*). The neem water extract utilized in this investigation contained various bioactive compounds, including phenolics, flavonoids, alkaloids, tannins, terpenoids, saponins, steroids, and cardiac glycosides. Flavonoids, terpenoids, polyphenols, alkaloids, phenolic acids, and other secondary metabolites have been identified as compounds that play a crucial role in the reduction of metal ions to zerovalent metals or in the stabilization of metal nanoparticles (MNPs). Isoflavonoids, flavonols, flavones, and flavanones represent a range of chemical classes. The capacity of flavonoids to form coordination complexes with metal ions is attributed to the presence of hydroxyl and carbonyl functional groups. The manufacture of metal oxide nanoparticles (NPs) can employ various amino acids, sugars, or fatty acids as readily accessible reducing and capping agents (*Sahu et al., 2016*). The findings of the study demonstrated that the ZnO-NPs, which were synthesized using a green method, exerted a significant impact on the viability of A549 and HCT 116 cells. Notably, this effect was observed even at the lowest doses of the samples administered. The results shown in this study exhibit a strong correlation with the conclusions reported by *Suresh et al. (2018)*, *Umamaheswari et al. (2021)*, *Naser et al. (2021)*, and *Selim et al. (2020)*.

In this study, the antibacterial activity of neem extract, ZnO-NPs, and green synthesized ZnO-NPs was evaluated against both gram-positive and gram-negative bacteria. The highest activity was recorded for green synthesized ZnO-NPs then ZnO-NPs, and finally neem extract. The present study recorded the phytochemical components of *A. indica*, including saponins, steroids, terpenes, tannins, glycosides, alkaloids, flavonoids, and phenols. These phytochemicals may account for the antibacterial activity of neem. Numerous studies have established a correlation between the existence of bioactive chemicals in plant materials and their antibacterial properties (*El-Mahmood, Ogbonna & Raji, 2010*; *Francine, Jeannette & Pierre, 2015*; *Itelima et al., 2016*). The antibacterial properties of ZnO-NPs can be ascribed to their direct contact with cellular walls, resulting in membrane distortion and rupture (*Brayner et al., 2006*; *Zhang et al., 2007*; *Sarwar et al., 2016*). The ability of ZnO-NPs to kill *Vibrio cholerae* bacteria and how they cause harm was studied (*Sarwar et al., 2016*). ZnO NPs interacted with *Vibrio cholerae* and made the membranes more fluid and depolarized. This led to protein leakage and changes in the structure of the *Vibrio cholerae* cells. It has been seen that bacteria treated with ZnO-NPs often have cell membrane damage, which is very toxic. The impact of ZnO-NPs on *E. coli* was investigated by *Leung et al. (2016)*, who observed that greater concentrations of ZnO-NPs resulted in the identification of cell damage sites. The damage seen in the cell membrane is caused by ZnO-NPs interacting with

bacterial membranes. There are bad effects on the molecular structure of phospholipids when certain factors interact with each other, which damages the cell membrane.

## CONCLUSIONS

The study found that green ZnO-NPs with non-spherical shapes, synthesized using neem aqueous extracts, had strong antibacterial activity against various Gram-positive and Gram-negative bacteria. In addition, the ZnO-NPs exhibited significant anticancer effects on HCT116 and A549 cancer cells. ZnO-NPs exhibit strong antibacterial and anticancer properties, making them promising candidates for novel therapeutic applications in biomedical fields.

### Funding

This work was supported by the Princess Nourah bint Abdulrahman University Researchers Supporting Project number (PNURSP2024R460), Princess Nourah bint Abdulrahman University, Riyadh, Saudi Arabia. This work was also supported by the Deanship of Scientific Research, Vice Presidency for Graduate Studies and Scientific Research, King Faisal University, Saudi Arabia (GRANT 5202). The funders had no role in study design, data collection and analysis, decision to publish, or preparation of the manuscript.

### Grant Disclosures

The following grant information was disclosed by the authors:
Princess Nourah bint Abdulrahman University Researchers Supporting: Project number (PNURSP2024R460).
Princess Nourah bint Abdulrahman University, Riyadh, Saudi Arabia.
Deanship of Scientific Research, Vice Presidency for Graduate Studies and Scientific Research, King Faisal University, Saudi Arabia: GRANT 5202.

### Competing Interests

The authors declare there are no competing interests.

### Author Contributions

- Hossam S. El-Beltagi conceived and designed the experiments, performed the experiments, analyzed the data, authored or reviewed drafts of the article, and approved the final draft.
- Marwa Ragab conceived and designed the experiments, analyzed the data, prepared figures and/or tables, and approved the final draft.
- Ali Osman conceived and designed the experiments, performed the experiments, analyzed the data, prepared figures and/or tables, authored or reviewed drafts of the article, and approved the final draft.
- Ragab A. El-Masry conceived and designed the experiments, analyzed the data, prepared figures and/or tables, and approved the final draft.

- Khairiah Mubarak Alwutayd performed the experiments, authored or reviewed drafts of the article, and approved the final draft.
- Hind Althagafi performed the experiments, analyzed the data, prepared figures and/or tables, authored or reviewed drafts of the article, conducted nanotechnology synthesis analysis, and approved the final draft.
- Leena S. Alqahtani performed the experiments, prepared figures and/or tables, and approved the final draft.
- Reem S. Alazragi performed the experiments, prepared figures and/or tables, and approved the final draft.
- Ahlam Saleh Alhajri performed the experiments, prepared figures and/or tables, and approved the final draft.
- Mahmoud M. El-Saber conceived and designed the experiments, analyzed the data, authored or reviewed drafts of the article, and approved the final draft.

## Data Availability

The raw data is available in the Supplemental File.

## Supplemental Information

Supplemental information for this article can be found online at http://dx.doi.org/10.7717/peerj.17588#supplemental-information.

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
