# Peer review of "Biosynthesis of zinc oxide nanoparticles via neem extract and their anticancer and antibacterial activities"

_PeerJ, doi:10.7717/peerj.17588_

## Round 0.1 · original submission · Major Revisions

It can be seen that the reviewers have raised different concerns over this study. Hence, there is a need to thoroughly revise this article in the light of the comments from the reviewers.

Reviewer 1 ·

Basic reporting

No comment

Experimental design

The authors should improve the ZnONPs sythentesis section in both chemical and green synthesis. They should indicate which chemicals were used as the reducing and capping agents of nanoparticles

Validity of the findings

Aurthors should compare their results with other findings for scientific validation of their study

Additional comments

See attachment.

Annotated reviews are not available for download in order to protect the identity of reviewers who chose to remain anonymous.

Reviewer 2 ·

Basic reporting

In this article, the authors describe the biosynthesis of zinc oxide nanoparticles from neem extract and attempt to screen them for anticancer and antibacterial activities. I appreciate the authors using different techniques in this study, which makes it interesting. However, several significant changes are needed before this article can be considered worthy of publication. Also, their claims of therapeutic properties are not convincing, and the authors must address various claims they make throughout the manuscript with solid scientific data.

1) The abstract section is written like results and discussion. Please look at other similar publications that do similar work and make sure that you revise your abstract section accordingly.

2) Avoid making vague statements just because you are citing another work. For example, in line 61, you say, "significant contributor to global mortality rates," but you do not provide any numbers or statistics.

- line 64 - several fatalities every year? How many? be specific

3) Paragraph 2 is not written in expert language and fails to consider several leading works in the field.
Several vague statements, too.

4) Line 78: What free radicals? Can you please specify?
Line 82: What are these antioxidants? Examples?
Line 92: 300 distinct bioactive chemicals? What are they? Consider adding a table, perhaps?
Line 94: What phytochemicals? What enzymes? There are so many vague statements again.
Line 101: Stable manner on a large scale? How? Why?
The authors must search, proofread, and identify other instances of these statements and fix them.
The examples that I have provided are not an exhaustive list.
Lines 103 and 104 need corrections as the statement does not make sense.
Lines 107, 108: Zinc oxide nanoparticles produced through eco-friendly methods are biocompatible and non-toxic - Why? What do eco-friendly methods have to do with biocompatibility or the NPs being non-toxic?

The entire introduction section needs a complete revision.

5) Figure 1 does not follow PeerJ's formatting. It is of very poor resolution, and the various texts are invisible.

Figure 2 has the same flaws.
Figure 3: In the abstract section, line 45, the authors claim 21 peaks. I do not see 21 peaks. The chromatogram is low resolution and is not labeled or annotated. The authors must fix this and include the original RW chromatograms in a supplementary file. How was integration performed on this? Please explain in the manuscript.
Figure 4: Please consider splitting this into two figures. Follow PeerJ guidelines.

None of the figures meet PeerJ requirements.

This manuscript, unfortunately, 'fails' concerning "basic reporting" in its present state.

Experimental design

6) Authors must ensure that the methods and materials used in this study are described with sufficient information to be reproducible by another investigator. In the present state, this is not the case.

Lines 136-138: Ensure you add product identifiers like product codes, etc.
Line 142: What is the dehydration procedure? How did the authors ensure that no contamination took place during this?
Line 143: No details on the mechanical blender; model? manufacturer? Settings used?
Line 144: How did you heat? Equipment details and experimental setup?
Line 145: What is the filter size? How many microns?
Line 146: What centrifuge? What temperature?
Line 146: How long was it stored in the fridge? Sample stability is very important - please specify how old your samples were at all the required sections and whether they were freshly prepared.
Line 152: Agitated how?
155: drop by drop over 45 minutes? Was this done manually, or did you have a setup?
158: How was the beaker sealed? 4 hours at what temp?

Also, the section does not mention the software used (for example, UV, DLS, ZETA), the version of the software used (example: ImageJ), the exact models of the instruments, and the exact set-up they used, for example, what kinds of cuvettes? Glass? Plastic? Quartz? What volume?

201: Mayer's reagent? Wagner's reagent? No details on the source.

Since there are many other similar points, I will leave it to the authors to identify the rest and fix them accordingly. Again, the list of examples I have provided is not exhaustive.

Clearly, 'fails' this section, unfortunately.

Validity of the findings

Line 498: Why haven't the authors done a Kirby-Bauer Test for the antibacterial activity? This test must be done for future submissions.

In Figure 8 on line 509, you say that "bacterial cells that had survived displayed diverse deformations," and yet you claim that these are anti-bacterial? Sounds contradictory. A Kirby-Bauer Test, as mentioned previously, would help you make this comparison better.

Line 530, 531: 21 distinct peaks? Your data does not show that.

The results and discussion section is written like a literature review. Please rewrite and make suitable changes.

Line 656, 657: Your work has failed to convince even remotely that this could be used to fight cancer. Please avoid making vague, unreasonable claims unless you have multiple results with robust data that support these. Cytotoxic substances are harmful not only to tumor cells but to all cells. In your work, you have not demonstrated a way to make these nanoparticles specific to cancer cells, and therefore, you cannot make claims like this.

The manuscript lacks concerning statistical analyses. For future submissions, the authors must perform the required statistical tests to demonstrate that the data are robust, statistically sound, and controlled.

The conclusion section is adequate. Repeats the results that the authors have already mentioned multiple times earlier.

'Fails' this section

Additional comments

Please revise the entire manuscript. Please look at relevant works that are similar, add experiments that are required, and present your data in a standardized manner that demonstrates how the data was acquired and exactly how you have drawn conclusions based on that robust data.

Also, what is the impact of ZnO nanoparticles on the environment and humans, especially from a therapeutic application perspective? The authors should cite relevant works that discuss these and add them in future submissions.

Reviewer 3 ·

Basic reporting

The manuscript by Hossam S. El-Beltagi and colleagues is well-written and presents a clear and concise overview of the research on the Biosynthesis of zinc oxide nanoparticles via neem extract, phytochemical screening for neem (Azadirachta indica A. Juss), and their anticancer, and antibacterial activities. Using chemical and green synthesis methods, the authors synthesized and characterized zinc oxide nanoparticles (ZnO-NPs). The authors further reported using ZnO-NPs synthesized by green methods as anticancer and antibacterial agents.

The authors have clearly stated the research question, conducted relevant experiments, and drawn reasonable conclusions. The manuscript is well-organized and easy to follow, but a few minor grammatical errors and awkward phrasings could be smoothed out.

The authors should be consistent with the nomenclature of the cell lines. They should use HCT 116 and A549 throughout the manuscript, including figures.

The font size in the following figure should be increased on the X and Y axis. Figure 1B-1D, Figure 2B-2D.

There is no scale bar in Figure 4.

The statistical significance should be included in Figure 6 and Figure 7.

Scale is not visible in Figure 8. Authors should write the scale bar in the figure legend.

Proofread the manuscript carefully for any grammatical errors or typos.

Ensure the references are complete and formatted according to the PeerJ style guide.

Experimental design

no comment

Validity of the findings

no comment

Additional comments

The discussion section could benefit from a more in-depth interpretation of the results and their potential implications in the context of existing knowledge.

Reviewer 4 ·

Basic reporting

No comment

Experimental design

The authors have designed series of biophysical and molecular-cell biology approach to compare and contrast the efficacies of chemically synthesized ZnO NPs, green ZnO NPs and neem extract as an anti-cancer and anti-bacterial agent.
Overall, the techniques used by the authors are relevant and the study has scope. However, for the ease of reading the authors can make minor edits in their manuscript presentation. For example, the figure legends can be more informative. Specifically, figure 4: The authors could mention the concentrations of NPs or neem extract used in analysis. In the methods section corresponding to figure 4, the authors have mentioned a wide concentration range but, the exact concentration of the reagent used for generating those images could be helpful.

Validity of the findings

no comments

---

## Round 0.2 · accepted · Accept

3 reviewers, including the most critical reviewer, are satisfied with the revision. The article can be Accepted

Reviewer 2 ·

Basic reporting

I am happy with the revisions. I recommend acceptance after minor revisions; the authors must thoroughly proofread to eliminate minor formatting and language errors.

Experimental design

NA

Validity of the findings

NA

Additional comments

NA

Reviewer 3 ·

Basic reporting

The authors have addressed my comments. I am satisfied with the changes.

Experimental design

No Comment

Validity of the findings

No Comment

Additional comments

No Comment

Reviewer 4 ·

Basic reporting

no comment

Experimental design

no comments

Validity of the findings

no comments

Additional comments

The authors have made necessary changes for the betterment of the manuscript. The changes made include clearer figure legends, detailed concentrations of the extracts and reagents used for the assays.